# A Genome-Scale Metabolic Model of *Methanoperedens nitroreducens*: Assessing Bioenergetics and Thermodynamic Feasibility

**DOI:** 10.3390/metabo12040314

**Published:** 2022-03-31

**Authors:** Bingqing He, Chen Cai, Tim McCubbin, Jorge Carrasco Muriel, Nikolaus Sonnenschein, Shihu Hu, Zhiguo Yuan, Esteban Marcellin

**Affiliations:** 1Australian Institute for Bioengineering and Nanotechnology (AIBN), The University of Queensland, Brisbane, QLD 4072, Australia; b.he@uq.edu.au (B.H.); t.mccubbin@uq.edu.au (T.M.); 2Australian Centre for Water and Environmental Biotechnology (ACWEB, Formerly AWMC), The University of Queensland, Brisbane, QLD 4072, Australia; chencai21@ustc.edu.cn (C.C.); s.hu@uq.edu.au (S.H.); z.yuan@uq.edu.au (Z.Y.); 3CAS Key Laboratory of Urban Pollutant Conversion, Department of Environmental Science and Engineering, University of Science and Technology of China, Hefei 230026, China; 4Department of Biotechnology and Biomedicine, Technical University of Denmark, 2800 Kongens Lyngby, Denmark; jcamu@biosustain.dtu.dk (J.C.M.); niso@dtu.dk (N.S.)

**Keywords:** genome-scale metabolic model, ANME archaea, reverse methanogenesis, bioenergetics, electron transfer, thermodynamic feasibility, MEMOTE

## Abstract

Methane is an abundant low-carbon fuel that provides a valuable energy resource, but it is also a potent greenhouse gas. Therefore, anaerobic oxidation of methane (AOM) is an essential process with central features in controlling the carbon cycle. *Candidatus* ‘Methanoperedens nitroreducens’ *(M. nitroreducens)* is a recently discovered methanotrophic archaeon capable of performing AOM via a reverse methanogenesis pathway utilizing nitrate as the terminal electron acceptor. Recently, reverse methanogenic pathways and energy metabolism among anaerobic methane-oxidizing archaea (ANME) have gained significant interest. However, the energetics and the mechanism for electron transport in nitrate-dependent AOM performed by *M. nitroreducens* is unclear. This paper presents a genome-scale metabolic model of *M. nitroreducens*, *i*MN22HE, which contains 813 reactions and 684 metabolites. The model describes its cellular metabolism and can quantitatively predict its growth phenotypes. The essentiality of the cytoplasmic heterodisulfide reductase HdrABC in the reverse methanogenesis pathway is examined by modeling the electron transfer direction and the specific energy-coupling mechanism. Furthermore, based on better understanding electron transport by modeling, a new energy transfer mechanism is suggested. The new mechanism involves reactions capable of driving the endergonic reactions in nitrate-dependent AOM, including the step reactions in reverse canonical methanogenesis and the novel electron-confurcating reaction HdrABC. The genome metabolic model not only provides an in silico tool for understanding the fundamental metabolism of ANME but also helps to better understand the reverse methanogenesis energetics and its thermodynamic feasibility.

## 1. Introduction

Methane is a major component of natural gas and a potent greenhouse gas. As a result, methane has significant economic value and environmental importance, particularly as global warming concerns escalate worldwide. The last few decades have seen increasing methane emissions, with methane concentration in the atmosphere increasing substantially since pre-industrial times, from 722 ppb to 1803 ppb [1]. In the global methane cycle, archaeal methane metabolism plays an important role in anoxic environments, with methanogenic archaea being the largest biological methane producers on Earth [2,3,4], while anaerobic methanotrophic archaea (ANME) reduce methane emissions by reversing methanogenesis, and thereby mitigating climate change [5,6].

The biological process of anaerobic oxidation of methane (AOM) was first discovered from a consortium of anaerobic methane-oxidizing archaea (ANME) and sulfate-reducing bacteria (SRB) in marine sediments [7,8,9]. Three distinct methanotrophic groups have been identified, which are ANME-1 (distantly related to *Methanomicrobials* and *Methanosarcinales*) [7], ANME-2 (related to *Methanosarcinales*) [10], and ANME-3 (related to *Methanococcoides* spp.) [11]. The three members of the ANME clades are often found to perform sulfate-dependent AOM [7,8,12,13,14]. Recent evidence has shown that AOM can also be coupled to metal oxides [15,16,17,18,19,20,21]. However, rather than transferring electrons to SRB or Fe (III), the member of the ANME-2d cluster, *Methanoperedens nitroreducens,* has been proven to utilize nitrates as the terminal electron acceptor [22,23,24] through a combination of metagenomics, metatranscriptomics and labeling experiments [23]. Currently, *M. nitroreducens*-like archaea have been identified in freshwater sediments [22,25,26], paddy soil [27], rivers and lakes [28,29]. Their prevalence suggests their importance in contributing to biological methane and nitrogen cycles in the anoxic environment.

From a thermodynamic point of view, the presence of energetically favorable electron acceptors are crucial to reverse the methanogenesis pathway. AOM is exergonic only when involving extracellular electron transfer or when direct interspecies electron transfer is involved. Sulfate-driven AOM has been shown to operate reverse methanogenesis at the limits of energetic feasibility (Gibbs free energy yields between −18 and −35 kJ/mol). In contrast, nitrate-dependent AOM is more energetically favorable with a Gibbs free energy change of −517 kJ/mol per methane. However, the growth rate of nitrate-AOM remains slow, with doubling times of weeks [22]. The slow growth suggests that other factors, rather than the overall thermodynamic driving forces, control the process of reverse methanogenesis and the growth rates [29]. Though the overall Gibbs free energy is negative, the first two reactions of reverse methanogenesis responsible for methane oxidation, the methyl-coenzyme M-reductase (Mcr) and N^5^-methyl-H_4_MPT: coenzyme Mmethyltransferase (Mtr), are endergonic under standard conditions and suspected to be the rate-limiting steps. The reversibility of the nickel-containing enzyme Mcr has been determined by kinetic parameters using purified Mcr from *Methanothermobacter marburgensis* [30]. Nevertheless, the thermodynamic challenge of operating the Na+-transporting Mtr in reverse methanogenesis has rarely been discussed.

Metagenomics and metatranscriptomics studies of ANME-1 [31], ANME-2a [32] and ANME-2d [23,24] support the same central pathway of methane oxidation in ANME archaea. However, their energy conservation mechanisms are slightly different, with specific terminal reductases and membrane-bound ion translocating enzymes in respiratory chains, which generate an electrochemical ion gradient and drive ATP synthesis. This general energy conservation framework enables only a shallow understanding of the energetic metabolism of ANME archaea. A detailed understanding of the various reactions and the mechanism involved in electron transfers remains to be elucidated. Current research suggests that the metabolism of *M. nitroreducens* is unique, which makes the energy metabolism of nitrate-coupled AOM more challenging to be understood. Strikingly, the ferredoxin-dependent respiratory enzyme complex, Rnf, which is typically essential in the methanogenesis pathway when Ech and Vho hydrogenases are absent, cannot be found in the genome. The Rnf complex is capable of transferring electrons from reduced ferredoxin to a membrane-soluble electron carrier using a sodium pump. As a result, it plays a significant role in the electron-transport chain and contributes to energy conservation, especially in methanotrophic pathways when the Na^+^ gradient is dissipated by Na^+^ pumping in Mtr. The novel cytoplasmic electron-confurcating heterodisulfide reductase HdrABC complex in *M. nitroreducens, i*MPEBLZ, is also a new characteristic of the metabolism of *Methanoperedens* species [24]. The HdrABC complex has been shown to catalyze a flavin-based electron bifurcation in the methanogenic pathway [33,34,35,36], whereas, in nitrate-driven reverse methanogenesis, HdrABC is proposed to form a reverse direction, which transfers electron pairs from reduced CoM-SH/CoB-SH thiols and ferredoxin to two oxidized F420 cofactors. This specific mechanism is thought to recycle ferredoxin instead of Rnf [24,29]. Whether this mechanism is associated with proton motive force remains unclear.

To better understand the metabolism of *M. nitroreducens* and the essential energy-conserving complexes in reverse methanogenesis, we constructed a genome-scale metabolic reconstruction of nitrate-driven AOM, which also serves as a functional annotation. Constraint-based metabolic models provide a framework to compute cellular functions and improve the understanding of specific metabolisms. Today, thousands of metabolic reconstructions are available for multiple organisms, including several methanogens [29,37,38]. However, a detailed systems-level characterization of ANME archaea is not available. In this work, we present a manually curated genome-scale metabolic model of *M. nitroreducens* adhering to current community standards in systems biology. This GEM is capable of providing an accurate quantitative estimate of electron transfer and bioenergetics in reverse methanogenesis. The model shows the essentiality of the electron-confurcating heterodisulfide reductase reaction, which is needed to balance redox and is also efficient in energy conservation. By describing the detailed electron flow from the cytoplasm to the membrane via each redox active complex, as well as by combining flux distribution predictions, the genome-scale metabolic model becomes a platform for understanding the thermodynamic feasibility of reverse methanogenesis. Finally, the model is used to show that coupling reactions are needed to achieve thermodynamic feasibility, and the energy transfer in redox cycles is important in conserving energy.

## 2. Results and Discussion

### 2.1. General Properties and Model Validation of iMN22HE

We present here the genome-scale metabolic reconstruction (GEM) of *M. nitroreducens,* named hereafter *i*MN22HE. The name follows the latest recommended conventions of model naming [39]. The lowercase “*i*” in italics represents *in silico*, and “MN” refers to the species indicator, *M. nitroreducens*. It then follows the iteration identifier, “HE”, for the primary model curators, published in 2022. A detailed description of how the model was generated and curated was provided in the Materials and Methods. The model contains 813 reactions, 684 distinct metabolites and 452 annotated genes, including the cytoplasmic internal reactions, transport reactions and exchange reactions. All the reactions were annotated with subsystems, with the most extensive subsystems being that of amino acid metabolism and vitamin and cofactor biosynthesis (Figure 1A). Due to the lack of information on the metabolism, the unusually high number of genes and metabolites with unknown functions suggest that a further investigation and potential refinement of the model may be needed in the foreseeable future.

The GEM *i*MN22HE was tested with MEMOTE (https://memote.io/, accessed on 12 January 2022) and SBML validator, which provides a platform for model quality testing [40]. The Memote total score is 83%, with a model consistency of 98.5%. The stoichiometric consistency, mass balance, charge balance and metabolite connectivity amount to 100%. After being cross-referenced by several different databases, the model also shows a better performance in annotation score (Figure 1B).

The model *i*MN22HE is saved in the SBML Level 3 Version 1 format. Additionally, the scripts used in the reconstruction and MEMOTE validation are publicly available through GitHub (https://github.com/computer-aided-biotech/iMN22HE, accessed on 8 February 2022).

### 2.2. Comparison of iMN22HE with Other Relative Models

The metabolic network described here was compared to other models of related metabolism (Table 1). The first manually curated genome-scale metabolic model of methanogenic archaea, *i*AF692, was constructed for *Methanosarcina barkeri* in 2006 [41]. Since then, multiple methanogen GEMs have become available, with the majority derived from *M. barkeri* and *M. acetivorans,* which synthesize methane from CO_2_/H_2_, formate, acetate, methylamines, methanol, or CO. The most noticeable difference between ANME archaea and methanogens is the reversed format of the methanogenesis pathway, which is the central pathway in ANME to catalyze AOM. Though the most recent *M. acetivorans* model, *i*MAC808, has been customized to capture a similar reverse acetoclastic pathway by co-utilizing methane and bicarbonate in the presence of suitable external electron acceptors, a flavin-based electron bifurcation HdrABC event was proposed in the model, showing a different electron transfer mode. The reduced ferredoxin generated by HdrABC was used in CO_2_ reduction, which contributed to the carbonyl group for acetate synthesis [35]. In *M. nitroreducens*, through the back reaction of HdrABC, CoM/CoB and reduced ferredoxin acted together as electron donors to reduce cofactor F420, which shaped a unique electron confurcation process. The reduced cofactor F420 then supplied reactants for the energy-conserving enzyme participating in the respiratory chain, F420H_2_: quinone oxidoreductase (Fqo), which collaborated with HdrABC on redox balancing of the essential cofactors F420, and ferredoxin [24]. Therefore, with the absence of membrane-bound ferredoxin-dependent respiratory enzymes, cytoplasmic HdrABC could realize energy conservation in reverse methanogenesis.

### 2.3. Model Prediction of Electron Confurcation Essentiality in Reverse Methanogenesis

In 2015, the soluble flavin-based electron confurcation HdrABC complex was first hypothesized to participate in reverse methanogenesis based on genomic and transcriptomic data and the redox potential analysis of these specific cofactors [24]. This mechanism theoretically satisfied the requirement of supplying heterodisulfide and oxidized ferredoxin for reverse methanogenesis, especially when the effective ferredoxin reoxidation enzyme Rnf was absent in the respiratory chain of *M. nitroreducens*. However, the HdrABC enzyme has only been assessed to perform electron bifurcation through GEMs of hydrogenotrophic methanogenesis [33] and acetotrophic pathways [34,35]; therefore, whether electron confurcation could meet the principle of mass balance and thermodynamic feasibility in the network and become the essential solution in reverse methanogenesis was required to be confirmed.

One of the premises that cofactor F420 could be involved in the reaction of heterodisulfide reductase was based on its complex structure. According to the structural model of the heterodisulfide reductase [NiFe]-hydrogenase complex (HdrABC-MvhAGD) from *Methanothermococcus thermolithotrophicus*, we knew that the key subunit HdrA containing flavi adenine dinucleotide catalyzed ferredoxin oxidation, HdrB performed CoM-SH/CoB-SH oxidation and HdrC with two [4Fe-4S] clusters provided channels for electron transfer [44]. In *M. nitroreducens* MPEBLZ, the cofactor F420 reduction interacted into the HdrABC complex as the adjacent location between cofactor F420 reducing hydrogenase (FrhB) and the HdrABC cluster [24]. To validate the coupling mechanism between Hdr and Frh, we replaced the electron-confurcating reaction to keep the Hdr function only:2 Fd *_oxidized_* + CoM-SH + CoB-SH = 2 Fd *_reduced_* + 2 H^+^ + CoM-S-S-CoB(R1)

However, under this variation, the model of *M. nitroreducens* was not able to grow. The incapability to grow was caused by the unbalanced ferredoxin redox cycle, as no oxidized ferredoxin generation mechanism existed in the network. In addition, the large potential difference between ferredoxin (E^0^′ = −520 mV) and CoM-SH and CoB-SH (E^0^′ = −143 mV) would cause waste in ∆G potential. If the electron confurcation was used only to overcome the ∆G barrier of heterodisulfide reductase, we could also consider using F420 reduction to drive CoM-S-S-CoB oxidation without confurcation.
F420 + CoM-SH + CoB-SH = F420H_2_ + CoM-S-S-CoB(R2)

Though it was more thermodynamically favorable, the model would not be stoichiometrically feasible due to ferredoxin imbalance. If a separate reaction of ferredoxin-F420 oxidoreductase (R3) was added, it could help. However, this reaction has not yet been identified.
2 Fd *_oxidized_* + F420 + 2 H^+^ = 2 Fd *_reduced_* + F420H_2_(R3)

This represents a true stoichiometric decoupling of two confurcation half-reactions. However, the ∆G of the half-reaction was highly positive (approximately 90 kJ/mol) and probably infeasible. Thereby, after comparing with the possible reactions in the Hdr-Frh complex, the electron confurcation was demonstrated to be an essential mechanism in the nitrate-driven AOM. Model simulations showed that the electron transfer direction was consistent with the hypothetical confurcating pattern, and the growth yield predicted by this condition was also in accordance with the experimental results shown in Haroon (2013) (Appendix A).

### 2.4. Bioenergetics Analysis of Steady-State Reverse Methanogenesis Using Flux Balance Analysis (FBA)

To further investigate the energetics of the metabolism, we performed flux balance analysis to simulate fluxes for the entire network. The model was constrained with experimental data from methane and nitrate uptake [23]. Simulations were performed by using the biomass equation and adenosine triphosphate (ATP) as the objective functions (Appendix A). Maximizing the ATP was used to elucidate energy conservation mechanisms in steady-state, and biomass was used to ensure the generation of all the necessary biomass precursors and predict growth. The growth rate of *M. nitroreducens* has been reported to be extremely low, and only about 1 percent of the consumed methane is utilized for biomass synthesis [45].

In anaerobic methane oxidation, the energy-producing process has already been suggested to be chemiosmotic coupling, as no other obvious oxidative phosphorylation methods in reverse methanogenesis were known. Hence, these membrane-bound electron transport enzymes were essential in connecting the whole process of nitrate-driven methane oxidation with the chemiosmotic energy conservation system. Their capability of translocating protons or sodium ions across the membrane constituted chemiosmotic potential and generated ATP synthesis. In the metabolic network (Figure 2), the apparent energy converting enzymes included the terminal nitrate reductase Nar complex, which, together with cytochromes, performed electron transfer from reduced menaquinone and translocated four protons out of the membrane. The Fqo complex pumped out up to three H^+^ while transferring electrons from menaquinone to the main redox cofactor F420. One more H^+^ was proposed to be driven by Fqo compared with F420H_2_: methanophenazine oxidoreductase (Fpo), as the higher potential difference between menaquinone (E^0^′ = −80 mV) and F420 (E^0^′ = −380 mV) than methanophenazine (E^0^′ = −165 mV) and F420 could raise the ion-pumping ability [24,46]. Besides, the protein machines methyl-transferring Mtr and membrane-bound heterodisulfide reductase HdrDE both performed the reverse direction, which sourced two sodium ions and two protons, respectively, from outside the cell while reversing methanogenesis. The established Na^+^ and H^+^ gradients were co-utilized by ATP synthase with the optimal Na^+^/H^+^ stoichiometric ratio updated in *Methanosarcina* [35], and a Na^+^/H^+^ antiporter Mrp assisted in optimizing the ATP synthase efficiency.

Using FBA, a quantitative prediction of flux distribution was obtained, which discovered important unknown activities of various enzymes in energy conservation. Prominently, the flux value of the membrane-bound HdrDE was much smaller than the cytoplasmic HdrABC complex in this reverse methanogenesis pathway (Figure 2, Appendix A) compared to other FBA predictions using GEMs for *Methanosarcinales* which also involved two classes of heterodisulfide reductase in the networks [34,35]. The split fluxes of these two complementary methods for generating CoM-S-S-CoB was obviously associated with the redox cycle of ferredoxin. With the absence of Rnf, HdrABC became the main supplier for oxidized ferredoxin, which performed the last step of reverse methanogenesis, formylmethanofuran dehydrogenase (Fmd). However, HdrDE still compensated a bit of CoM-S-S-CoB, as ferredoxin could be re-oxidized through acetyl-CoA synthetic reaction (Cdh) and other biosynthetic processes in the reconstruction network. Thus, considering the ferredoxin redox balance, HdrABC must carry a predominant flux due to the high flux ratio of Fdred/Fdoxi in the metabolism.

However, the primary cause of HdrABC dominance was not the lack of Rnf, but the difference in bioenergetic efficiencies. In fact, HdrABC could even give an energy-conserving advantage over the Rnf complex in nitrate-driven AOM. To compare the energetic efficiencies of HdrABC and Rnf, under the assumption that the cell seeks to maximize energy production under a steady state, the energy production mechanism was simulated by maximizing ATP synthesis (Figure 3, Appendix A).

Within the electron-confurcating HdrABC, the demand of CoM-S-S-CoB and Fd*_oxi_* for methane oxidation was both supplied by cofactor F420 reduction. In the F420 redox cycle, cytoplasmic HdrABC and the two successive F420-dependent steps in central reverse methanogenesis both generated high flux F420H_2_ according to the model, except for the F420-dependent NADP reductase (F4NR), which catalyzed a small amount of cofactor F420 oxidation with NADP^+^ for anabolic activities. The majority of reoxidation was processed by Fqo, accompanied by 13.2 protons being pumped out of the cell. With 17.6 protons translocated through the Nar/cytochrome complex and 2.2 Na^+^ pumped into the cell catalyzed by Mtr, an ion gradient of 28.6 H^+^/Na^+^ was driven by this process. The HdrABC available nitrate-driven AOM was predicted to yield 7.15 ATP theoretically via the model simulation. However, the alternative hypothesis with a pseudo-Rnf reaction instead of HdrABC was suggested to form a different energy-conserving mechanism. As the potential difference between ferredoxin (E^0^′ = −520 mV) and menaquinone (E^0^′ = −80 mV) was large, the Rnf complex was supposed to be an efficient enzyme translocating four sodium ions across the membrane when catalyzing ferredoxin oxidation by menaquinone. Under the hypothesis of the ∆*hdrABC* mutant, our model suggested that the membrane-bound HdrDE carried all the CoM-S-S-CoB oxidation activity, which caused 2.2 H^+^ to be pumped into the cell, resulting in a directly opposite effect on ATP synthesis. The flux value of Fqo also decreased under the ∆ *hdrABC* mutant, as no new cofactor F420 could be reduced via CoM-S-S-CoB and ferredoxin oxidation, which resulted in a lower proton gradient. The Nar/cytochrome complex and Mtr enzyme translocated the same amount of protons, as the redox balance in the quinone pool was constant. Thereby, though the sodium-pumping Rnf was responsible for energy conservation and oxidized ferredoxin regeneration, this whole hypothetical process could only gain 23.2 H^+^/Na^+^ ion gradient and was predicted to generate 5.5 ATP. Approximately one less ATP was synthesized than HdrABC variant metabolism. Therefore, even if the Rnf is available in the network for Fdoxi regeneration, the metabolism was still predicted to carry flux through HdrABC (Appendix A) because of the higher bioenergetic efficiency. Therefore, though cytoplasmic HdrABC could not generate membrane potential directly, the coupling mechanism of efficient energy conservation made the electron-confurcating HdrABC a significant contributor to the energetics of nitrate-driven AOM.

### 2.5. Electron Transfer during Nitrate-Driven Methane Oxidation in M. nitroredencens

The model enabled the electron flow from methane oxidation to the terminal electron acceptor nitrate to be depicted with unprecedented details by understanding the flux distribution. For example, the model quantitatively presented how the eight electrons from methane were serially received by the three cytoplasmic redox-active cofactors, CoM-S-S-CoB, oxidized F420 and oxidized ferredoxin, and produced fully oxidized product carbon dioxide. Channelled by their respective redox loops, the electrons from the cytoplasm were accepted by the membrane-bound electron-translocating energy conservation enzymes in the network. After entering the membrane-soluble menaquinone pool, electrons could finally be shifted to the Nar/cytochrome complex and reached nitrate located at the extracellular side of the membrane. Through the Nar complex, nitrate was reduced to nitrite by accepting two electrons. As a result, the balanced electron transfer mode of 4 mol nitrate oxidizing 1 mol methane was consistent with the experimental data [23] and the model predictions. The presence of nitrite reductase (Nrf) in metagenomes has shown the possibility of further reducing nitrite to ammonia by *M. nitroreducens* itself [24]. However, in normal conditions with abundant nitrate supply, there was no energetic benefit and, therefore, no predicted flux through the Nrf enzyme. With the different reducing strengths and electron transfer abilities between Nar and Nrf, the activity of Nrf was assumed to be associated with the electron acceptor’s availability.

The electron movements in the cytoplasm before electrons approached the membrane-bound electron transport system were also poorly understood, especially for the routes linking CoM-S-S-CoB oxidoreductase. In nitrate-driven reverse methanogenesis, the two heterodisulfide reductases both operated in the direction of oxidizing CoM-SH/CoB-SH to CoM-S-S-CoB, with the concomitant transfer of two electrons. However, when CoM-S-S-CoB acted as an oxidant to drive methane oxidation, we found that the reduced products CoB-SH and CoM-SH were not released by a single reaction, but produced by Mcr and Mtr, respectively, which are the first two sequential reactions in reverse methanogenesis. A similar coenzyme-releasing mode of CoM-S-S-CoB has been demonstrated by soaking experiments showing that the CoM-S-S-CoB structure was clamped between two noncubanes (4Fe-4S) and homolytically cleaved. According to the structural data, the catalytic mechanism of CoM-S-S-CoB reduction occurred via a one-by-one electron transfer to successively release CoB-SH and CoM-SH [44]. Consequently, we speculated that the CoM-S-S-CoB reduction in reverse methanogenesis also followed the same “one-by-one” electron transfer mechanism, showing the same coenzyme releasing order with a balanced electron transfer amount. Therefore, when considering the whole process of CoM-S-S-CoB oxidoreductase, Mcr and Mtr were regarded as a chain reaction in the reverse methanogenesis pathway as well.

In the process of CoM-SH/CoB-SH oxidation, the two heterodisulfide reductase classes not only generated two energetic mechanisms but also resulted in two different electron transfer modes. In the HdrDE-centred electron transfer route, the electrons released from Mcr and Mtr, which flow via coenzyme B and coenzyme M redox loops, were directly transferred to the in-membrane electron carrier, menaquinone, via membrane-bound HdrDE. Subsequently, the electrons transmitted by a quinone-loop could be accepted by nitrate (Figure 4A). The cytoplasmic HdrABC, as described previously, was coupled with the energy-conserving Fqo through a cytoplasmic F420 redox loop. Thereby, the electrons transferred by HdrABC had to pass an F420-loop and a quinone-loop to finally accomplish the oxidoreductase process, and the routes are shown in Figure 4B.

### 2.6. Thermodynamic Feasibility in Endergonic Methane Oxidation

To reverse the canonical methanogenic pathway, AOM must be thermodynamically feasible. This is achieved by coupling suitable terminal electron acceptors such as nitrate, sulfate, or iron to achieve thermodynamic feasibility. When nitrate is the only terminal electron acceptor driving AOM, the change in the Gibbs free energy is exergonic (∆*G* = −503 kJ mol^−1^) [23]. However, a few reactions are endergonic under the standard conditions, such as Mcr, Mtr, and the newly-added electron-confurcating reaction HdrABC (Appendix A). While in nature, this is not uncommon, in AOM, the energy transfer process remains to be clearly illustrated. For metabolism to function, each reaction needs to be thermodynamically feasible under cellular conditions. This can be achieved by carefully balancing metabolites concentration that ensures the lowest free energy change for all the reactions in the pathway or driven by other exergonic reactions in the energy coupling mechanism.

To assess the thermodynamic feasibility of the electron flow, we explored options for the AOM electron transfer chain in *M. nitroreducens,* as illustrated in Figure 4. By tracing electron transfer routes centered around the redox cofactor CoM-S-S-CoB, we suggested that the first two electrons lost from methane have to go through a series of redox loops to be finally accepted by nitrate. All reactions were linked together by electron transfer and integrated into a complete redox cycle, which comprised the reactions Mcr, Mtr, HdrDE, HdrABC, Fqo, and Nar (as shown in Figure 4). Through this loop, potential energy can be transferred through an electron shift in a redox process. The stoichiometry of the redox cycle and the ∆*G* of the reactions are shown in Appendix A. Using the flux values predicted by the model simulation (Appendix A), in which HdrDE carried (0.183/1.1) and HdrABC carried (0.917/1.1) of the total flux, we showed that HdrDE and HdrABC share the load of electron translocation. Then, we calculated the thermodynamic feasibilities of the redox cycle under standard conditions (Appendix A).
∆*G_redox cycle_* = ∆*G_Mcr_* + ∆*G_Mtr_* + (0.183/1.1) ∆*G_HdrDE_* + (0.183/1.1) ∆*G_Nar_* + (0.917*/*1.1) ∆*G_HdrABC_* + (0.917*/*1.1) ∆*G_Fqo_* + (0.917*/*1.1) ∆*G_Nar_* = 37.751 kJ mol^−1^.

The negative ∆*G* value shows that, though the initial process of oxidizing methane to methyl-H_4_MPT in AOM was endergonic, the reduction of nitrate was sufficient to drive the endergonic reactions. We speculate that the endergonic methane oxidation reactions (Mcr and Mtr) in other methanotrophic archaea might follow a similar thermodynamic path. To further analyze the thermodynamic feasibility of reverse methanogenesis, we suggest the use of a thermodynamic MFA (tMFA) [47].

## 3. Materials and Methods

### 3.1. Metabolic Model Reconstruction

The process of metabolic network reconstruction thoroughly followed the main steps listed in Thiele and Palsson (2010) [48], we began with genome annotation based on the whole genome sequence of *Candidatus Methanoperedens nitroreducens* strain ANME-2d from NCBI (NCBI Reference Sequence: NZ_ JMIY01000000). The genome sequence was annotated using the RAST server on default settings [49], and the draft reconstruction could be automatically created by using Kbase (https://kbase.us/, accessed on 11 July 2020) [50] and ModelSEED (https://modelseed.org/, accessed on 11 July 2020) [51]. The draft model provided a preliminary framework of cellular metabolism, significantly improving the efficiency of reconstructing a new metabolic model. During model curation, we manually refined and expanded the model on a pathway-by-pathway basis based on the biochemical information about ANME-2d and relative organisms available in the literature, as well as the public databases, including Kyoto Encyclopedia of Genes and Genomes (KEGG) [52], Bio-Cyc [53], BRENDA [54] and Uniprot [55]. The directionality of reactions was generally determined with Gibbs free energy calculations and also checked with the information from BiGG [56] and literature sources. The gene-protein-reaction (GPR) assignments were constructed by Kbase and manually refined based on literature and gene annotation. Gap-filling was also performed manually by finding the metabolite dead-ends in the model and mapping reactions to KEGG and MetaCyc pathway information. Reaction and metabolite nomenclature, as well as subsystems and EC numbers, were all consistent with BiGG databases.

Finally, the genome-scale metabolic model of *M. nitroreducens* ANME-2d was constructed, including 684 unique metabolites and 813 metabolic reactions. A complete table containing all the reactions, metabolites and biomass compositions is available in Appendix A and can be found in sbml and json format in Appendix A. The scripts for model reconstruction and MEMOTE validation can be accessed from the following GitHub repository: https://github.com/computer-aided-biotech/iMN22HE (accessed on 8 February 2022).

### 3.2. Model Simulation with Flux Balance Analysis

The model simulation was performed using flux balance analysis (FBA), an established technique applied to predict phenotypes for metabolism reconstruction [57]. Reactions and their participating metabolites in the *M. nitroreducens* metabolic network were connected by the stoichiometric matrix *S* (m*n), where m is the number of metabolites and n is the number of reactions. In FBA, the cellular system was assumed to be under pseudo-steady-state growth and could be represented by the equation:*S* × *v* = 0
in which *v* (n*1) is a vector of reaction flux. Upper and lower bounds of flux through individual reaction were imposed as additional constraints as follows:*V_i, lower_* ≤ *V_i_* ≤ *V_i, upper_*

To find feasible flux distributions that simulate the likely physiological conditions, FBA used linear programming (LP), which is subject to mass balance and flux constraints dealing with an optimization problem. In this paper, the model growth was maximized by maximizing both ATP demand and biomass production. The biomass equation formulation was adopted from the *i*MB745 model [34], and limiting substrate uptake rates were set based on data available in the literature [23]. The growth and non-growth-associated ATP maintenance parameters were set to be 169.9 mmol ATP per g of cell mass and 5.0 mmol ATP per gram of cell mass h^−1^, based on the previous model of *Methanococcus maripaludis i*MR539 [33], as the low growth rate predicted by the model was comparatively closed to ANME archaea. FBA optimization problems were solved by the Gurobi optimizer (https://www.gurobi.com/, accessed on 9 October 2020) using COBRA Toolbox 3.0 [58] in Matlab 2018b (Mathworks Inc., Natick, MA, USA). The genome-scale metabolic network was visualized using the Escher software [59]; a map of all reconstructed metabolic pathways is presented in Appendix A.

### 3.3. Thermodynamic Calculations

The standard Gibbs free energy of each reaction in methanogenesis, including the canonical reverse methanogenesis and electron transport chain, was calculated based on the published standard midpoint potentials of redox couples in standard conditions (Appendix A). The equation we used is:∆*G* = −*nF* ∆*E*

*F* is the Faraday constant, 0.09648 kJ/eV, and *n* is the number of moles of electrons transferred in the reaction. The Gibbs free energy of the reactions in the reverse methanogenesis pathway in *M. nitroreducens* are presented in Appendix A.

## Figures and Tables

**Figure 1 metabolites-12-00314-f001:**
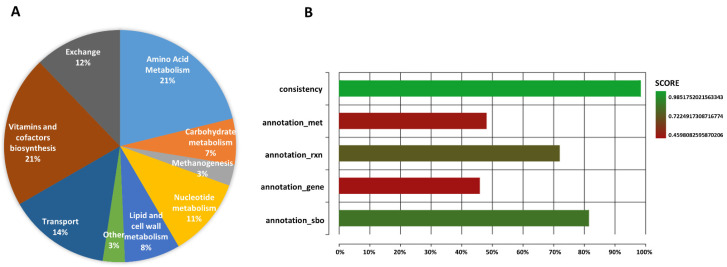
(**A**) The distribution of reactions in *i*MN22HE sorted by metabolic subsystems. (**B**) MEMOTE score of iMN22HE regarding the quality of model consistency, metabolites, reaction, gene and sbo term annotation degree. The quality of the GEM reconstruction is the total evaluated with 83%.

**Figure 2 metabolites-12-00314-f002:**
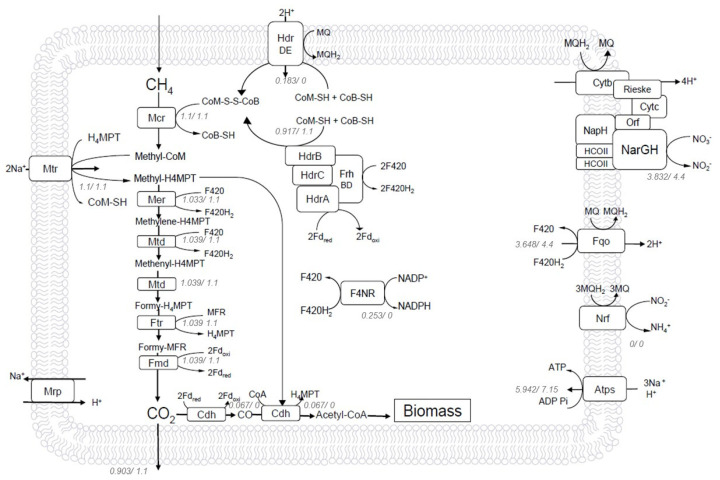
Reverse methanogenesis pathway supported by the model *i*MN22HE of *M. nitroreducens*. As shown, methane is the only carbon source. Through the reverse methanogenesis pathway, most of it is oxidized to carbon dioxide, with a small amount of carbon flux through the reductive acetyl-CoA pathway to generate acetyl-CoA for cell growth. Numbers in *italics* next to enzymes denote reaction fluxes (mmol gDW^−1^h^−1^) calculated under biomass (**left**) and ATP (**right**) maximum. Nitrate is the terminal electron acceptor, reduced by the NarGH-Rieske/cytochrome *b* complex, which drives the reverse methanogenesis in *M. nitroreducens*. Mcr, methyl-coenzyme M reductase, HdrABC, soluble F420-dependent heterodisulfide reductase, Mtr, methyl-H4MPT: coenzyme M methyltransferase, Cdh, CO dehydrogenase, Fqo, membrane-bound F420H2: quinone oxidoreductase, Nrf, nitrite reductase, F4NR, F420-dependent NADP reductase.

**Figure 3 metabolites-12-00314-f003:**
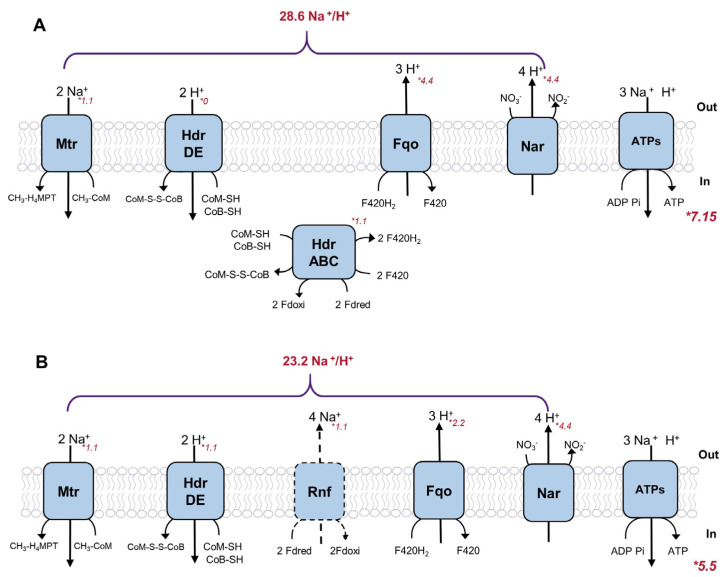
All ion-translocating enzymes of M. nitroreducens in action and the amount of ion translocation. (**A**) Normal energy-conserving related enzymes in nitrate-driven AOM. (**B**) An alternative hypothetical respiratory chain with a pseudo-Rnf reaction (in dashed line) instead of HdrABC. Numbers in italics next to enzymes denote reaction fluxes (mmol gDW^−1^h^−1^) calculated under ATP demand.

**Figure 4 metabolites-12-00314-f004:**
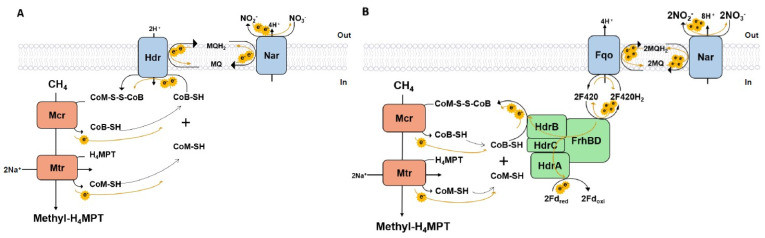
Electron transfer routes related to two heterodisulfide reductase classes. (**A**) Electron tranfer routes centred on membrane−bound heterodisulfide, HdrDE. (**B**) Electron transfer routes centred on in-cytoplasmic heterodisulfide, HdrABC. The yellow arrows represent the electron movements, and the dashed arrows correspond to the hypothetical electron−confurcating mode.

**Table 1 metabolites-12-00314-t001:** Properties comparison of *i*MN22HE with related methanogen metabolic reconstruction.

Organism	Model	Mets	Rxns	Central Metabolic Pathway	Main Energy-Conserving Enzymes	Soluble Heterodisufide (HdrABC)	Citations
*M. nitroreducens* *M. barkeri*	*i*MN22HE *i*AF698	684 558	813 619	Reverse methanogenesis Hydrogenotrophic methano-genesis;	Fqo Fpo Ech Vho	Electron confurcation NR	[41]
				Methylotrophic methanogenesis			
	*i*MG746	718	815	Hydrogenotrophic methanogenesis;	Fpo Ech Vho	Electron bifurcation	[42]
				Methylotrophic methanogenesis			
	*i*VS941	708	705	Acetoclastic methanogenesis; Methylotrophic methanogenesis	Fpo Rnf	NR	[43]
*M. acetivorans*	*i*MB745	715	818	Acetoclastic methanogenesis; Methylotrophic methanogenesis	Fpo Rnf	Electron bifurcation	[34]
	*i*MAC868	707	839	Acetoclastic methanogenesis; Methylotrophic methanogenesis;	Fpo Rnf	Electron bifurcation	[35]
*M. maripaludis*	*i*MR539	605	570	Reverse methanogenesis Hydrogenotrophic methano-genesis	Eha/Ehb	Electron bifurcation	[33]

NR, not reported; Mets, metabolites; Rxns, reactions; Fqo, F420H_2_: quinone oxidoreductase; Fpo, F420H_2_: phenazine oxidoreductase; Ech, ferredoxin-dependenthydrogenase; Vho, methanophenazine-dependent hydrogenase; Rnf, methanophenazinereductase; Eha/Ehb, energy-conserving hydrogenases.

## Data Availability

The model files, reconstruction scripts and relevant data can be found at https://github.com/computer-aided-biotech/iMN22HE (accessed on 8 February 2022).

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
