# Peer review of "A Genome-Scale Metabolic Model of *Methanoperedens nitroreducens*: Assessing Bioenergetics and Thermodynamic Feasibility"

_metabolites, 2022, doi:10.3390/metabo12040314_

Round 1

Reviewer 1 Report

The manuscript describes a genome-scale metabolic model of Methanoperedens nitroreducens, a methanotrophic archaeon that performs anaerobic oxidation of methane via a reverse methanogenesis pathway and utilizes nitrate as the terminal electron acceptor. The manuscript is based solely on in-silico tools with sound methods and reliable results. The work is undeniably significant as it helps understand reverse methagenesis, a process that removes a potent greenhouse gas from the atmosphere. 

Author Response

We than the reviewer for his/her positive comments regarding our manuscript. We thank you for considering our work "undeniably significant as it helps understand reverse methagenesis, a process that removes a potent greenhouse gas from the atmosphere".

Reviewer 2 Report

The author developed a genome-scale metabolic model of Methanoperedens nitroreducens and used it to assess bioenergetics and thermodynamic feasibility of reverse methanogenesis. I think this is a very nice work and the manuscript is well-written. I only have a couple of minor suggestions.

Line 82, iMPEBLZ -> MPEBLZ

Line 357, I suggest ‘thermodynamics-based metabolic flux analysis (tMFA)’ and including a reference.

Author Response

We thank the reviewer for his/her positive comments. We have addresses teh changes as follow:

Line 82, iMPEBLZ -> MPEBLZ

Changed iMPEBLZ to MPEBLZ.

Line 357, I suggest ‘thermodynamics-based metabolic flux analysis (tMFA)’ and including a reference.

The reference has been added. 

Reviewer 3 Report

Thank you for the opportunity to review this work. The manuscript under this review discusses a genome-scale metabolic reconstruction (GEM) for understanding the thermodynamic feasibility of reverse methanogenesis, and the energy transfer in redox cycles is important in conserving energy.

The study needs several concerns that need to be addressed before the manuscript could be improved for publication in Metabolites.

Abstract

The abstract must be re-worded and not present full stops. Please, justify the text.

Introduction

Well-presented and written, but they should increase the number of references that support the theoretical framework presented and improved several concerns, especially in the methane definition (first paragraph).

Results and Discussion

2.1. General properties and model validation of iMN22HE

The description of the model must be taken to Material and Methods, where it must be described in detail. Line 114-120.

More generic descriptions of the procedures and pertinent references are needed, this can then be followed by identification of specific equipment brands and manufacturers (usually inserted parenthetically).

2.2. Comparison of iMN22HE with other relative models

Table 1 or Figure 1? Please, change this aspect. Line 134.

If it is table 1 presented on later pages, please change it at the end of the paragraph. Figure 1, in this case, is not referenced in the text. Please indicate figure location.

2.4. Bioenergetics analysis of steady-state reverse methanogenesis using flux balance analysis (FBA)

The first time you present ATP in the text, indicate the meaning of the acronym. Line 200.

Where is figure 2? Line 234.

Change figure 3 to the end of the paragraph where it is presented. Line 277.

This paragraph is very long and difficult to read. I suggest the authors to add full stops to facilitate reading. Lines 242-277.

In this section I do not find the conclusions reached by the authors. Please indicate them.

Material and Methods

The Material and Methods section is incomplete, and I suggest the authors add information; it does not correctly describe how the model was generated and several supplementary material figures should be moved to this section to explain it correctly.

It needs an improvement; it does not provide enough information to understand how the methodology has been carried out and it is difficult for the reader to have an idea.

I suggest adding the URLs of the databases used. Lines 368-369.

Author Response

Reviewer 3 requested several revisions to enhance the manuscript quality, we hope to have thoroughly addressed, and revised the manuscript could be acceptable for publication. Here, reviewer comments are written in black and answers to those comments in blue.

Abstract

The abstract must be re-worded and not present full stops. Please, justify the text.

 As suggested, the abstract has been re-written

Introduction

Well-presented and written, but they should increase the number of references that support the theoretical framework presented and improved several concerns, especially in the methane definition (first paragraph).

We thank the reviewer for the suggestion, more references are cited in the new introduction.

Results and Discussion

2.1. General properties and model validation of iMN22HE

The description of the model must be taken to Material and Methods, where it must be described in detail. Line 114-120.

More generic descriptions of the procedures and pertinent references are needed, this can then be followed by identification of specific equipment brands and manufacturers (usually inserted parenthetically).

Part of the model description was removed and included in the Material and Methods, with more details about the model description. Line 114-118 is reserved in this section, as text is relative to Figure 1A.

More descriptions and references have been added to explain what was done. The software versions and URLs are also added in this section and Materials & Methods section.

2.2. Comparison of iMN22HE with other relative models

Table 1 or Figure 1? Please, change this aspect. Line 134.

If it is table 1 presented on later pages, please change it at the end of the paragraph. Figure 1, in this case, is not referenced in the text. Please indicate figure location.

Thank you for pointing this out. Tables and figures are now correctly referenced. Table 1 is very large and utilises a full page. Hence has been appended.

2.4. Bioenergetics analysis of steady-state reverse methanogenesis using flux balance analysis (FBA)

The first time you present ATP in the text, indicate the meaning of the acronym. Line 200.

Thank you.  Adenosine triphosphate (ATP) instead of ATP has been added.

Where is figure 2? Line 234.

Change figure 3 to the end of the paragraph where it is presented. Line 277.

This paragraph is very long and difficult to read. I suggest the authors to add full stops to facilitate reading. Lines 242-277.

The Figure 2 and Figure 3 are relocated to where the context are presented, giving a better structure of the text and figures. The paragraph has been  restructured.  

In this section I do not find the conclusions reached by the authors. Please indicate them.

In this section, we quantitatively described the relationship between ATP synthase and bioenergetics relative enzymes, including membrane-bound enzymes and cytoplasmic enzyme HdrABC, indicating that cytoplasmic HdrABC also contributed the bioenergetics of M. nitroreducens. This conclusion has been added.

Material and Methods

The Material and Methods section is incomplete, and I suggest the authors add information; it does not correctly describe how the model was generated and several supplementary material figures should be moved to this section to explain it correctly.

It needs an improvement; it does not provide enough information to understand how the methodology has been carried out and it is difficult for the reader to have an idea.

I suggest adding the URLs of the databases used. Lines 368-369.

As suggested, we added more information about metabolic model reconstruction and model simulation with FBA.

Round 2

Reviewer 3 Report

I thank the authors for improving the manuscript with the recommended considerations.

Congratulations on the work and I wish you the best in the future.

Best regards.
